# Research of the Smart City Concept in Romanian Cities

Simona Bălășescu [1], Nicoleta Andreea Neacșu [1], Anca Madar [1], Alexandra Zamfirache [2,*]
and Marius Bălășescu [1]

1   Department of Marketing, Tourism Services and International Business, Transilvania University of Brasov, 500084 Brasov, Romania
2   Department of Management and Economic Informatics, Transilvania University of Brasov, 500084 Brasov, Romania
*   Correspondence: alexandra.zamfirache@unitbv.ro

**Abstract:** The Smart City concept has emerged in the last decade as a fusion of ideas about how information and communication technologies could improve the functioning of cities. A new concept, that of a sustainable Smart City, is already under discussion. This article aims at analyzing the Smart City concept in Romania. The resulting advantages, but also the difficulties and obstacles that need to be confronted, are considered when becoming a Smart City. When a city wants to become smart, it must consider both the advantages and the difficulties it will face in this endeavor. This paper has been able to take into account and group the four key features of a sustainable Smart City. The authors conducted research in two parts. The first consisted of conducting a comparative analysis of the most important results of Smart City projects implemented in the four reference cities in Romania compared to London. The second, a quantitative analysis, aimed to analyze the opinions and attitudes of Romanians regarding the Smart City concept in relation to sustainability. An important finding of the study shows that over half of the respondents are familiar with the Smart City concept and 41.9% consider health as a priority for investment in technology. The authors of the article propose clearer highlighting and division of cities from the point of view of creating a Smart City.

**Keywords:** Smart City; sustainable development; smart transportation; smart governance; smart waste management; quantitative research; consumer attitudes





## 1. Introduction

In the 2018 UN Report on the prospects of urbanization of the world [1] it is estimated that 68% of the world's population will live in urban areas by 2050 and that, in many areas, the share of the population living in the city and the number and size of cities will increase [2]. The rapid pace of urbanization and the unplanned expansion of cities bring about important changes to economies at all levels. If the consumption of water, fuels and electricity are taken into discussion, growing pollution with a strong impact on the lives of citizens and the environment can already be imagined. Although only 2% of the planet's surface is covered by cities, they consume 80% of the total energy produced worldwide and produce 75% of the total carbon dioxide emissions [3].

The high and rapid level of urbanization requires new and innovative ways of managing the complexity of urban living (problems caused by overcrowding, energy consumption, resource management and environmental protection) [4]. According to the United Nations (2018) [5], urbanization is closely connected to the three dimensions of sustainable development: economic, societal and environmental. Well-managed urbanization can help maximize the benefits of congestion while reducing environmental degradation and other potentially negative effects of an increasing number of urban dwellers.

Cities are very dynamic entities that rely on the continuous flow of people, ideas, resources and, in general, the connections they have with other entities in the areas they coordinate. In order to prosper, cities must respond to the economic and social needs of

their inhabitants. They need to effectively manage their impact on the environment in order to ensure sustainable and durable growth to the benefit of all.

There are a number of reasons that force cities to take action in order to become "smart": demographic and environmental factors, vulnerability to natural disasters, inability of infrastructure to cope with rapid social and economic changes, as well as economic and financial pressures (the need "to do more with less") as a result of increased competition between cities.

The implementation of projects related to the transformation of cities into Smart Cities is seen as a tool for the development of a sustainable society. They must integrate all sustainability perspectives, including social, environmental and institutional aspects, and cultural pillars [6].

This article aims to: (1) analyze the implementation stage of a Smart City in Romania, (2) identify the advantages of such a city and (3) identify the risks and obstacles that confront the city when becoming a Smart City. In this context, the analysis considers the sustainable development and functioning of a Smart City. The Organisation for Economic Cooperation and Development (OECD) defines sustainability as "the extent to which the net benefits of the intervention continue, or are likely to continue" [7]. Sustainable development is a model of economic growth in which the use of resources aims to meet human needs, while preserving the environment, so that these needs can be met not only now but for future generations [8]. In order to achieve sustainable global development, in 2015 the leaders and Governments of the United Nations member states agreed on Agenda 2030, which includes 17 Sustainable Development Goals (SDGs) together with 169 associated targets, with the aim of being fully implemented by 2030 [9].

The reference sources were accessed from the following databases: ScienceDirect, Springerlink Journals, Proquest Central and Clarivate Analytics Web of Science, which were accessed through Transilvania University in Brașov. The information about the smart projects of the cities analyzed in the paper was obtained by accessing the websites of their town halls. The search keywords were: smart city, sustainability, renewable energy, waste management, smart governance and smart transportation. Through this search, the authors found a total of 426 works from which they selected those presented in the References section according to the relevance and novelty of the information contained and the year of publication.

To achieve the purpose of the paper, the authors conducted research in two parts. The first consisted of conducting a comparative analysis of the most important results of Smart City projects implemented in the four reference cities in Romania compared to London (this being representative of a Smart City). The authors have analyzed the Smart City projects implemented in four emblematic cities in Romania: Brașov, Bucharest, Cluj and Sibiu, which are in the top 10 in terms of the number of Smart Projects implemented. In addition to this research, the authors made another, quantitative in nature, to identify the opinions, attitudes and perceptions of citizens (from the four cities analyzed above) regarding the Smart City concept. In the specialized literature, different authors have identified several key features of smart cities. After analyzing the literature, the authors took into account four key features of a sustainable Smart City: smart transportation, smart governance, renewable energy and smart waste management. One of these features is smart waste management. From that, at the end of the quantitative research, the authors used the chi-square test to check whether there was a link between the gender of respondents and their habit of selecting household waste. The authors chose to test this feature because, in this type of project, citizens are directly involved and the connection between the variables could be highlighted more.

The value of this paper lies in the analysis of the concept from both parties: what the authorities did to become a Smart City and the point of view of consumers. Additionally, the contribution brought by this paper, shows where Romania is, at present, from this point of view. The paper shows that the barriers to Smart City development primarily consist of a lack of information and lacking education of the population. Thus, we propose the

public authorities carry out, as soon as possible, programs of information, education and awareness for the Romanian population regarding the broad Smart City concept.

This work is structured in five sections: Introduction, Literature Review, Research Methodology, Results and Discussion and Conclusions.

The Introduction presents the background of the theme (the trend of population evolution and its migration to the urban environment and the negative implications of the agglomeration of cities and the need for a smart planning of their activity). In this section, the purpose of the work, and the methodology used to achieve it, are also presented.

In the Literature Review section, the authors synthesized the information presented in the studied articles and grouped them according to the four key characteristics.

The Research Methodology section presents the two types of research that were carried out in the work (a comparative analysis and a quantitative one) and establishes the main objectives of the research based on the four key features of a Smart City.

In the Results and Discussions section, the results of the two types of research are presented. These were analyzed in comparison to the results of other studies, which can be found in the articles presented in the References section.

The Conclusions section summarizes the most important ideas and results of the paper and includes the authors' proposal to calculate an economic and sustainability index to measure how smart a city is.

The results of the research can be used to define public policies by regional and local authorities, as well as to define the adoption of a system of indicators (as proposed in this work) that measures how intelligent a city is.

## 2. Literature Review

### 2.1. Developing the Concept of Sustainability in Smart Cities

The guiding principle of sustainable development is meeting human needs while protecting the current and future availability of resources [10]. Mensah defines sustainability as an entity's ability to maintain itself over time. Other approaches in the scientific literature have defined the concept as a dynamic of equilibrium between the satisfaction of human needs and environmental protection; this is in line with the contributions of the Brundtland Report [11].

Across the globe, every city dreams of becoming a Smart City. Integration of information and communication technologies to optimize the lifestyle of people is the process involved in developing a Smart City. Each government thinks about and takes various measures to fulfill the necessities of becoming a Smart City [12].

The world's urban population is expected to exceed six billion by 2045. Much of the projected urban growth will take place in countries in developing regions, especially in Africa. Consequently, these countries will face many challenges in meeting the needs of their growing urban population, including housing, infrastructure, transport, energy and employment, as well as basic services such as education and health care [1].

Franchina et al., (2021) [13] point out that there is still a weak connection between smart and sustainable urban practices, despite the potential of ICT to improve green living. They define a sustainable city that can be described as an urban agglomeration whose main objective is to contribute to improvements in quality and environmental protection, to social equity and well-being and to economic performance in the long term. The same idea is found in Ahvenniemi et al., (2017) [14], which mentions that many of the smart solutions for Smart Cities are not aligned with sustainability objectives, thus generating the concept of sustainable Smart Cities. As stated by Prince Antwi-Afari et al., (2021) [15], creating smart and sustainable cities from the beginning can help solve urbanization problems and can lead to sustainable development.

A Smart City must pay attention to the needs of its people, rational resource management, sustainable development and economic sustainability [16]. A city with more efficient services and more sustainable and environmentally smarter energy use is required [17]. The same authors have developed a method, called Smartainability, to help decision makers

understand and quantify the potential benefits of implementing innovative technologies that enable smart services for cities. Additionally, conducting a ranking of cities is an important tool that can help cities understand their performance in different dimensions of urban sustainability, compared to other cities in the same region, and identify areas for improvement [18].

### 2.2. Analysis of the Smart City Concept

The Smart City concept emerged during the last decade as a fusion of ideas about how information and communications technologies might improve the functioning of cities, enhancing their efficiency, improving their competitiveness and providing new ways in which problems of poverty, social deprivation and poor environment might be addressed [19].

A Smart City will have to be one that provides efficient services, has good mobility, ensures safety and security, has a good image, is sustainable and bases these factors on economic development [20]. Smart cities bring together technology, government and society to enable a smart economy and smart mobility, environment, people, living and governance [21].

The Smart City type of urban market was valued at USD 739.78 billion in 2020 and is estimated to reach USD 2036.10 billion in 2026, with an annual growth rate of 18.22% between 2021 and 2026 [22]. This market is influenced by the increasing share of urban populations in the total population of the world. Although there has been more and more talk in recent years about the development of Smart Cities, it is difficult to find a unanimously accepted definition. Caragliu et al., (2009) [23] argue that "a city is smart when the investments in the human and social capital, in the traditional transport and modern communications (ICT) infrastructure contribute to the sustainable economic development and to a superior quality of life, with a wise management of natural resources, by participatory governance" [24]. Another definition [25] states that a Smart City is a place where the traditional services and networks are made more efficient by using telecommunications and digital technologies to the benefit of its citizens and economy.

Grossi and Trunova [26] emphasize that a Smart City must be technological, interconnected, sustainable, comfortable and safe.

The British Standards Institute (BSI) defines a "Smart City" as an "efficient integration of physical, digital and human systems in order to build the necessary background for the sustainable, prosperous and inclusive development of the future of its citizens" [3].

The smart development of a city aims to increase the quality of life of the citizens living in that city by reducing poverty, unemployment and, similarly, by efficient management of energy resources. The "Smart City" concept emerged at the end of the last century as a necessity for effective management of the rapid development of cities which highlighted the importance of ICT (Information and Communication Technology) in their development. By this concept, a connection between citizens and public management is made, as they are no longer perceived as simple users or public services consumers, but as partners in the development of the city. A Smart City assumes the existence of an integrated informatics system that includes a multitude of cloud computing subsystems, Internet of Things (IoT) devices, Open Data, Big Data and mobile applications connected to the Internet through secure networks. These allow the local administration to interact directly with citizens and the infrastructure of the city [27].

This paper, based on study of the literature, identified several elements of a Smart City.

The authors consulted 426 specialized articles from different databases (ScienceDirect, Springerlink Journals, Proquest Central, Clarivate Analytics Web of Science), from which 96 were selected according to the relevance and novelty of the information contained and the year of publication.

From the specialized analysis, several key features of a Smart City emerged, of which four stood out. They are shown in Figure 1. Based on these four key characteristics, the authors built objectives for the quantitative research.

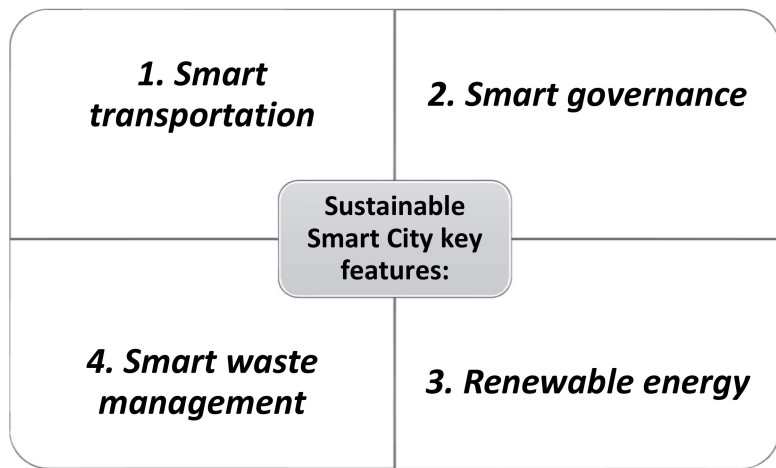

**Figure 1.** The key features of a sustainable Smart City.

### 2.2.1. Smart Transportation

The rapid increase in the amount of traffic in cities has led to both an increase in traffic jams and accidents and to increases in pollution and time spent in traffic. Hence the need for "intelligent mobility" in cities, through the use of public transport, in combination with the personal car or instead of it [28]. In a Smart City, the goal is the mobility of citizens—the ability to connect them in an efficient and elegant way with their points of interest—and not just their transportation from one point to another on the city map [3,29].

According to Matei et al., (2018) [30], for a city to become smart, a first step is to have smooth and decent traffic and, for that, the authorities must get involved. A European Commission study by DG Communications Networks, Content and Technology [31] states that a sustainable urban mobility plan is needed to implement the latest ITS/ICT solutions. The same study stated the three pillars of sustainable mobility: more fluid, safer and more accessible. Sustainable forms of transport are a key issue for cities across the globe, including smart cities [32,33].

Kanthavel et al., (2021) [34] define the intelligent transport system as a set of information and communications technologies used in transport systems to improve the safety, efficiency and stability of the transport network, as well as to reduce traffic congestion and improve the experience of drivers.

Intelligent transport, as claimed by Benevolo et al., (2016) [35], must take into account the following elements: public transport management to optimize public transport line networks, passenger information so that they can decide on the appropriate mode of transport, safety in vehicles with camera systems for greater protection for passengers, electronic payment systems to allow passengers to purchase tickets via the Internet, and comfort [36].

Cepeliauskaite et al., (2021) [37] states that, in the transport sector, actions are aimed at demotorizing and decarbonizing transport, including the expansion of electric vehicles.

The literature describes several solutions by which a city can transform its transportation system into a smart one. These include: the expansion of electric vehicles; greater use of energy-efficient appliances and autonomous and shared/public mobility systems [37]; dynamic trip-planning and ticketing services, on-demand minibuses and first- and last-mile ridesharing [38]; and walking and cycling [36], among others.

### 2.2.2. Smart Governance

The Smart Economy aims to digitize the economy, from payments, payments by phone and electronic wallets to the use of NFC or QR technologies. The goal of a Smart Economy is to finally digitize the entire experience of the consumer, and possibly to employ the exclusive use of electronic currency (cryptocurrency) [39]. Better use of the citizens'

expertise will also require better trained managers and leaders. This new category of professionals will need to know how to work, speak and decide with the citizens [40].

Smart governance, or e-governance, aims to inform and involve citizens in the life of the city, to reduce bureaucracy and to increase the transparency of governance. Gil-Garcia et al., (2016) [41] state that citizen-centered means the administration of a Smart City is oriented towards public value. Exercising power in a Smart City implies the need to give "visibility" to specific government issues [42]. Bolivar-Meijer [43] highlights six attributes of a smart governance system, which must be based on ICT, external collaboration and participation, internal coordination, the decision-making process, e-government and results. Accessing information about public institutions via the Internet, filling in online forms, e-commerce [44,45], paying taxes online [46], submitting documents to the National Agency for Fiscal Administration (NAFA) via the Internet and using e-mail or electronically signed documents in relation to public institutions become increasingly normal ways of interacting with public administration [47,48]. In order to effectively coordinate the many departments and to have access to real-time data, public administrations need intelligent systems and tools. Smart solutions, such as web portals, online forums, mobile applications and their integrated services, ensure two-way communication between authorities and citizens and help the latter voice their questions, suggestions and dissatisfactions. A United Nations [49] document states that the design of electronic platforms through electronic participation must take into account three levels, namely electronic decision making, electronic consultation and electronic information. According to Lim-Yigitcanlar [50], e-participation, which is part of e-governance, includes three main elements: e-information, e-consultation and e-decision-making. In Italy, thanks to the implementation of the Digital Citizenship Card, the Publi Digital Identity System, the Electronic Identity Card, the National Service Card and the National Register of Resident Population, citizens and businesses can access public services online using a single pair of credentials [48].

### 2.2.3. Renewable Energy

Smart solutions for the environment are the optimization of energy, water and electricity consumption, air quality monitoring and waste management. It has become imperative to make cities smarter in order to handle large-scale urbanization and to find new ways to manage energy and improve living standards without harming the environment in cities [51]. Lewandowska et al., (2020) [52] state that energy infrastructure stands out as one of the key elements of a Smart City, especially since its state and structure determine whether the principles of sustainable development will be implemented.

According to Gesteira-Uche (2022) [53], in 2016 the European Commission developed guidelines for the promotion of near-zero energy buildings, as the construction sector is responsible for almost 40% of global energy consumption and over 30% of greenhouse gas emissions [54]. Haas et al., (2021) [55] say that it is important to study and improve energy efficiency in order to promote global and coordinated development of the economy, society and the environment. He also points out that there are three major areas in the energy system to invest in: renewable energy (RE) technologies, electricity grid infrastructure and reducing energy demand while increasing energy efficiency in buildings. Liu et al., (2022) [56] say that energy efficiency not only emphasizes the relationship between a country's economic growth and energy consumption, but can effectively reflect a country's level of green development.

Building automation systems can reduce GHG (greenhouse gas) emissions, better air quality can be achieved as a secondary benefit of many energy saving and mobility applications, and the detection and control of water losses can support the conservation of water resources [57]. Small and micro renewable energy installations are used in multi and single family residential buildings that contain various devices, and also this type of facilities can power street lamps, road signs, vehicles and parking meters [52]. Additionally, lighting systems represented by smart light-emitting diodes (LEDs) [58], smart urban

mobility, smart grids and smart metering [59] are all viable solutions for securing the energy of a Smart City.

Other alternative sources for renewable energy that can be used in a Smart City in order to reduce pollution are: solar energy, wind energy, hydropower and oceanic energy, biomass energy and geothermal energy [60]. However, as Kanase-Patil et al., (2020) [51] and Alizadeh et al., (2016) [61] explain, due to their intermediate nature their use is still restricted. The integration of renewable energy sources in the electricity system of cities would determine the solution to problems such as pollution, climate change and dependence on fossil fuels [62].

2.2.4. Smart Waste Management

As DelufaTuzJerin et al., (2022) [63] point out, in fast-growing cities in developing countries, managing waste in an environmentally acceptable way is a major challenge. Various waste management issues have become more difficult to solve in recent decades. Waste management includes collection, transport, processing and disposal [64]. Related to this area, there are problems with collection services, landfill location and reverse logistics applications [65]. The European Union has introduced the three Rs waste management program—reduce waste, reuse, recycle—so that there is no more waste buried [66]. In order to meet the challenges of the environmental impact of waste, it is necessary to move from a linear economy to a circular economy [67]. The selection of waste for domestic and industrial consumers, the construction of ecological landfills for waste and the capture and reuse of thermal energy resulting from their decomposition are some modern methods of management [68,69]. Other measures implemented in smart waste management are: development of a smart waste management system which helps local authorities to monitor the waste collection contractor by bringing information about garbage container status and automatically reporting when it is full [70,71]; a landfill monitoring process using wireless sensor networks (WSN) [72] which use NodeMCU (web-based IoT solution) and an ultrasonic sensor to create a wireless prototype device for real-time monitoring of trash levels [73]; and designing optimal waste transport routes to reduce the negative impact of these activities on the environment [74].

Table 1 summarizes key features of a Smart City that were identified by the authors based on analysis of the literature.

**Table 1.** Key features of a Smart City.

| Topic | Selected Sources |
| --- | --- |
| Smart transportation | Xin Li et al., (2017); |
| | Vrabie-Dumitrașcu (2019);<br>Han Zhang et al., (2022);<br>Matei et al., (2018);<br>DG Communications Networks, Content and Technology (2012);<br>Behrendt (2019);<br>Garau et al., (2016);<br>Kanthavel et al., (2021);<br>Benevolo et al., (2016);<br>Ribeiro et al., (2021);<br>Cepeliauskaite et al., (2021);<br>Canales et al., (2017). |

**Table 1.** *Cont.*

| Topic | Selected Sources |
|---|---|
| Smart governance | Romanian Association for Smart City (2021a); |
| | Noveck (2015);<br>Gil-Garcia et al., (2016);<br>Argento et al., (2019);<br>Bolivar-Meijer (2016);<br>Alghazzawi-Badri (2022);<br>Ribeiro-Duthie et al., (2021);<br>Tyutyuryukov-Guseva (2021);<br>Ministry of Communications and Information Society (2016);<br>Battilani et al., (2022);<br>United Nations (2020);<br>Lim- Yigitcanlar (2022). |
| Renewable energy | Kanase-Patil et al., (2020); |
| | Lewandowska et al., (2020);<br>Gesteira și Uche (2022);<br>European Commission (2016);<br>Haas et al., (2021);<br>Liu et al., (2022);<br>Romanian Association for Smart City (2021b);<br>Strielkowski et al., (2020);<br>Komninos (2022);<br>Bibri (2020);<br>Alizadeh et al., (2016);<br>Al-Nory (2019). |
| Smart waste management | Delufa Tuz Jerin et al., (2022); |
| | Batur et al., (2020);<br>Medeiros Assef et al., (2022);<br>Popov-Kuzmina (2021);<br>Viswanathan-Telukdarie (2022);<br>Xu-Yang (2022);<br>Peura et al., (2022);<br>Omar et al., (2016);<br>Misra et al., (2018);<br>Longhi et al., (2012);<br>Muniandy et al., (2018);<br>Hariyani1-Meidiana (2018). |

## 3. Research Methodology

To achieve the purpose of the paper, the authors conducted two types of research. The first consisted of conducting a comparative analysis of the most important results of Smart City projects implemented in the four reference cities in Romania compared to London (this being representative of a Smart City). The authors have analyzed the Smart City projects implemented in four emblematic cities in Romania. In addition to this research, the authors conducted another analysis, quantitative in nature, to identify the opinions, attitudes and perceptions of citizens (from the four cities analyzed above) regarding the Smart City concept.

The quantitative research is an applied descriptive one which aims to analyze the opinions and attitudes of Romanians regarding the Smart City concept in relation to sustainability point. Considering the above aspects, the authors considered it important to know the opinions of citizens from four very important cities in Romania regarding the Smart City concept. This research can help public authorities make certain decisions in order to improve the citizens' quality of life.

The general hypotheses of the quantitative research were the following:

I1. Most of the respondents know the Smart City concept;
I2. A large part of the respondents do not use renewable energy sources;
I3. Most subjects collect waste selectively.

The main objectives of the research are based on the four key features of a Smart City that have been identified by the authors:

O1. Determining *smart transportation* practices, including public and private transportation;
O2. Identifying *smart governance* practices, including online administrative services provided by public institutions and companies;
O3. Finding out the most important actions related to *renewable energy*;
O4. Identifying *smart waste management actions*.

The quantitative research was conducted using a survey and a questionnaire was used as a data collection tool. The questionnaire used in the research was distributed in an electronic format by means of a web-based platform. This data collection technique is called Computer Assisted Web Interviewing (CAWI) [75] and is a method where the questions in the questionnaire are displayed on a web page and the respondent only needs to fill in the answers directly in the browser page. The data collection period was between February and April 2022.

The researched population consisted of the inhabitants of four representative cities in Romania, namely Brașov, Bucharest, Cluj and Sibiu. These cities were selected by the authors because their local authorities are concerned with the implementation of projects that will lead to their transformation into a Smart City. In Romania, these four cities are in the top 10 in terms of the number of Smart Projects implemented.

The sampling method used in this research is called "snowball". It is a non-probability sampling technique that does not allow extrapolation of results. The purpose of this sampling method is to get as many people as possible recruited into the sample. In this sense, the link leading to the web page of the questionnaire was distributed on social networks (Facebook, WhatsApp, Instagram) and the users of these networks were asked to access the questionnaire and answer its questions. Two eligibility criteria were taken into account: being over 18 years old and living in one of the four analyzed cities (Brașov, Bucharest, Cluj and Sibiu).

## 4. Results and Discussion

*4.1. Comparative Analysis of Implementing the Smart City Concept in Emblematic Cities in Romania*

In Romania, the Smart City concept is relatively new. Although Bucharest has been a pioneer since 2007, when it implemented a traffic management system, many Smart City projects have been carried out in the country.

Romania has 265 towns, but the groups of professionals working in the Smart City field are few and located only around larger cities, representing attempts to become a Smart City [76].

According to Vegacomp Consulting [77], the major challenge for today's Romanian cities is the speed of project implementation while overcoming bureaucracy and changing mentalities in all media, both public and private. The same source notes that, currently, 594 Smart City projects are being implemented and this market is valued at over EUR 120 million, which demonstrates and confirms substantial growth of the Smart City market in Romania.

When analyzing all Smart City projects in Romania, it is noticed that the leader is smart mobility, with 188 projects, followed by smart governance, with 130 projects. The rostrum is completed by smart living with 121 projects, smart economy with 84 projects, smart environment with 42 projects and smart people with 29 initiatives [77].

The authors have analyzed the Smart City projects implemented in Brașov, Bucharest, Cluj and Sibiu (Table 2).

**Table 2.** Smart City projects in Cluj, Brașov, Bucharest and Sibiu.

| No | City | Overall Projects | Smart Economy | Smart Mobility | Smart Environment | Smart People | Smart Living | Smart Governance |
|----|------|------------------|---------------|----------------|-------------------|--------------|--------------|------------------|
| 1 | Cluj-Napoca | 54 | 4 | 20 | 7 | 3 | 12 | 8 |
| 2 | Brașov | 18 | 1 | 6 | 1 | 1 | 3 | 6 |
| 3 | Bucharest District 4 | 18 | 2 | 7 | 0 | 0 | 5 | 4 |
| 4 | Sibiu | 16 | 5 | 5 | 0 | 0 | 2 | 4 |

Source: Vegacomp Consulting [77].

The authors analyzed the available information about the four cities in Romania regarding the Smart City projects implemented and compared them with those about London (UK)—which is considered as a Smart City reference. The results of the comparative analysis, presented below, are structured on the four objectives of the quantitative research:

- All four cities in Romania analyzed, compared to London, have implemented many projects in the field of *public and private transport*, designed to ensure both traffic flow and pollution reduction in cities. These projects have materialized in: computerized traffic management (Brașov, Bucharest, Cluj from Romania and London, UK), integrated infrastructure for bicycle and pedestrian traffic (Cluj, Brașov, Sibiu, London), vending machines and public transport season tickets (Brașov, Bucharest, London), public transport prioritization systems (Brașov, Cluj, London), online platforms through which citizens obtain information about the traffic situation in the city (reporting traffic problems) (Bucharest), online or SMS parking payment systems (Brașov, Bucharest, Cluj, London), charging stations for electric cars (Brașov, Bucharest, Cluj, Sibiu, London), intelligent traffic light systems for crowded intersections (Brașov, London), purchase of electric public transport (Brașov, Bucharest, Cluj, Sibiu, London) and bicycle/scooter rental systems to encourage people to use less green transport (Brașov, Bucharest, Cluj, Sibiu, London);
- Regarding the *administrative services* offered by public institutions and other companies in the analyzed cities, projects such as electronic administration services (Brașov, Bucharest, Cluj, Sibiu, London) and electronic geospatial services, which serve as a public information portal (Brașov, Bucharest, Cluj, Sibiu, London), were implemented;
- For the use of *green energy*, solar panels were installed to illuminate pedestrian crossings during the night (Brașov, London);
- In the field of *selective waste collection*, projects have been implemented that have integrated waste management systems (Brașov, London) and developed selective waste collection programs (Brașov, London).

What has also been done in London, and could be done in Romanian cities, includes: a map of solar panel installation opportunities to help companies and other organizations identify opportunities to install solar panels on their property; a project for the exchange of solutions, practices, experiences and results with other cities in Europe and the improvement of the way cities manage and share data, which aims to create better and more energy-efficient living conditions for citizens and communities in London; GovTech—a platform to help innovative SMEs better understand the opportunities that exist in London and provide them with tools to help secure these contracts; and the Civic Innovation Challenge which aims to bring together London's public sector and private organizations with innovative technology companies trying to solve some of the most pressing problems in London.

As can be seen, the cities analyzed have taken important steps on their way to Smart City status. Objectives that lead to this status more quickly and completely must continue to be pursued.

Any newly implemented activity presents benefits and risks. For the Romanian cities analyzed in this paper, the benefits of the transition to Smart City status identified by the authors are: reduction of $CO_2$ emissions and improvement of air quality; streamlined traffic and a reduction in the amount of noise produced; lower usage of personal cars for moving

around the city; increased transparency and ethics in the provision of services by local public authorities through efficient management; a reduction in the amount of time it takes to solve citizens' problems while avoiding trips to the headquarters of local authorities by digitizing their activities; increased quality of services provided by local authorities; decreased energy losses from buildings by insulating them and using alternative energies for heating which decreases their heating/cooling expenses; early detection of water losses on the network which avoids increased expenses; a reduction in the amount of waste that must be stored (lack of space) and an increased degree of waste recycling which will lead to a reduction in the amount of raw materials used; and better management of waste collection.

The identified risks and obstacles of the transition to being a Smart City include the following: not all citizens (especially the elderly) will be able to adapt to the new measures (digitalization of services, waste selection); the implemented measures will cause price increases for some services, which could displease some citizens; the failure of implemented software would cause blockages to current activity (technology dependence); most of these projects were realized with funds from the European Union and the procedures for granting these funds are very cumbersome, which leads to delays in the implementation of the projects; and delays in receiving installments of money which also causes delays in the realization of projects. Some of these risks are also found in the work of Razmjoo et al., (2021) [78].

Although there are considerable risks in this activity, the authors believe that the advantages of a Smart City are indisputable, and the risks can be assessed and monitored.

The comparative analysis above aims to show the current situation regarding the implementation of the Smart City concept in our country. Starting from the results of the comparative analysis, the authors wanted to find out what the opinions of Romanian citizens are regarding the Smart City concept. For this purpose, quantitative research was conducted.

### 4.2. Opinions, Attitudes and Perceptions of Romanian Citizens Regarding the Smart City Concept

The research started with the questioning the respondents about the Smart City concept, and the data collected showed that 46% of respondents knew of this concept while 54% did not. This leads to the conclusion that more than half of the respondents have not heard about the Smart City concept and, consequently, they do not know what it means.

The previous analysis showed that there are four main sectors that need to be developed for a city to become smart:

- Types of personal transport, a public transport system and intelligent and green transport infrastructure;
- Online administrative services provided by public institutions and companies;
- Renewable energy;
- Selective waste collection.

The quantitative research conducted aimed to identify the opinions, attitudes and perceptions of Romanian citizens regarding these sectors. The research results are structured around the four main objectives.

The sample structure is given in Table 3.

**Table 3.** The sample structure.

| Criteria | Sample Structure | | | | | |
|---|---|---|---|---|---|---|
| | **Sample (1116 Respondents)** | **49% Men** | | **51% Women** | | |
| | **Residents** of | Brașov 31% | Cluj-Napoca 31% | Sibiu 20% | Bucharest 18% | |
| | Sample structure according to **age** | Under 20 2% | 20–29 years 29% | 30–39 years 40% | 40–49 years 18% | 50–59 years 9% | 60–69 years 2% |

*O1. Determining **smart transportation** practices, including public and private transportation*

The answers received regarding the means of transportation that the respondents use most frequently show that 61.6% of the respondents use their personal car, 30.8% use a public means of transportation, while 3.2% use a bicycle. A total of 3.2% use another means of transportation and 1.1% use a motorcycle. It can be concluded that most of the respondents, more than half (62%), use the most their personal car as a means of transportation.

A good example for the increased degree of attractiveness of public transportation for citizens is put into effect by the European Union, which has implemented the CIVITAS Initiative (an action that supports cities in implementing a sustainable, clean and efficient integrated transport policy from the point of view of energy). CIVITAS II is an initiative by which 14 cities have implemented measures aimed at increasing the quality of urban public transport [79].

More than half of the respondents (58%) occasionally use public transportation. A total of 23% of the respondents use public transportation several times a week, while only 16% of them use it daily (Table 4).

**Table 4.** The frequency with which respondents use public transport.

| | Frequency | Percent | Valid Percent | Cumulative Percent |
|---|---|---|---|---|
| Daily | 184 | 16.5 | 16.5 | 16.5 |
| A few times a week | 256 | 22.9 | 22.9 | 39.4 |
| Occasionally | 648 | 58.1 | 58.1 | 97.5 |
| Others | 28 | 2.5 | 2.5 | 100 |
| Total | 1116 | 100 | 100 | |

This research aimed to establish the degree of satisfaction respondents had for public transportation in the city. The intermediate response variant (neither satisfied nor dissatisfied) accounted for 47% of responses. Almost a quarter of the subjects (30.1%) are fully satisfied with public transport, while 23% of them are dissatisfied. The frequency distribution indicates a relatively balanced concentration of responses on the positive and negative side of the scale (B2B score = 23% and T2B score = 30.1% positive).

From the respondents' point of view, the main problems that public transportation companies should solve are:

- Poor cleaning of buses;
- Operating hours;
- Reduced number of buses at peak hours;
- Absence of electric buses;
- Absence of a lane dedicated to public transport.

According to a survey conducted by PwC among Europeans, 74% of consumers will opt for the most convenient way to travel, including the use of several types of transportation, and 28% of European vehicle owners believe they can earn money from sharing their cars in a peer-to-peer platform. However, the global 2020 Digital Auto Report shows that those data have changed, and the global COVID-19 Pandemic has changed consumer choices in this area as well, with most choosing to use their own car instead of public transportation or the mobility platform [80].

Among the subjects, 80% personally own a car and 20% do not have a car. If we compare the data obtained in this research with data from Europe, according to the PwC Digital Auto Report, Europe's car fleet will increase by 1.4% by 2025 [81], reaching approximately 273 million of cars. It is estimated that the number will then decrease by 5.4% by 2030 due to the development of mobility platforms (such as Uber, Clever, Bolt, etc.) and alternative ownership methods [81]. Globally, alternative mobility models will account for between 17% and 28% of global vehicle transport by 2030.

The research continued with identification of the type of engine in the car that the respondents own. Only the subjects who answered *Yes* to the previous question were invited to answer this question, namely, those who own a personal car. For this reason, only 896 responses were given out of a total of 1116 respondents.

Half of the respondents (49%) own a car that runs on petrol, and close to 46% of the respondents drive a car that runs on diesel. A small percentage have a hybrid car (3%), 1% have an electric car. 1% own a car which does not belong to any of these categories.

From the preferences of consumers on the international market, it is noted that petrol engines continue to be the most sought after in Germany and the USA, while in China 68% of consumers under the age of 40 prefer electric engines, which compares to only 46% in Germany and 37% in the USA. The forecasts show that battery electric vehicles (BEVs) will account for 17% of new vehicle sales in the European Union and 19% in China by 2025. In the USA, they will have a share of only 5% by 2025, due to reduced government support [81].

The identification of conditions in which the respondents are willing to ride a bike is highlighted the following results:

- 65.8% would ride a bicycle if there were more bicycle renting points;
- 56.1% would ride a bicycle if drivers were more civilized;
- 27.7% would ride a bicycle if there were more bike lanes;
- 12.2% had other responses.

*O2. Identifying* **smart governance** *practices, including online administrative services provided by public institutions and companies*

The research results showed that 84% of the respondents use online services pro-vided by public utilities institutions and companies, while 16% of them do not use them. The increased percentage shows the desire of the population to use online services as much as possible (including for receiving and paying bills, paying taxes, online schedules, etc.). Of course, the global COVID-19 pandemic has left its mark on these results, as the recommendations referring to responsible social conduct [82] advocate avoiding direct contact.

The respondents were further asked to express their degree of satisfaction degree regarding online services provided by public utilities institutions and companies. The obtained results are shown in Figure 2.

The most frequent answers are offered at levels three and four of the scale. A total of 39.1% of the respondents awarded level three, while 32.3% of the respondents indicated rank four. The fewest answers, representing 4.9%, are indicated by those who are totally dissatisfied with the online services offered by public institutions and public utility companies. At the other end of the spectrum, only 12% of the respondents indicated that they are very satisfied with these services.

The answer variants for scales four and five are combined, representing the Top Two Box (T2B) score which received 44.3% of the provided answers. At the other end of the spectrum, the score called Bottom Two Box (B2B) totals 16.6%. Comparing the answers received in the research, it is noted that 44.3% of the respondents are satisfied with the online services offered by the public institutions and the public utility companies.

A proposal called E-Romania has already been launched. This is a public policy in the field of e-governance within the Administrative Capacity Operational Program (POCA) [83]. Through the implementation of this program, the capacity of Romanian institutions and public authorities to develop and implement e-government solutions (representing a series of important services in the life of citizens and private legal entities) has increased. Thus, it is expected that, in the coming period, investments by public institutions in various online applications will increase, and with this increase it becomes imperiously necessary to deepen research in this field.

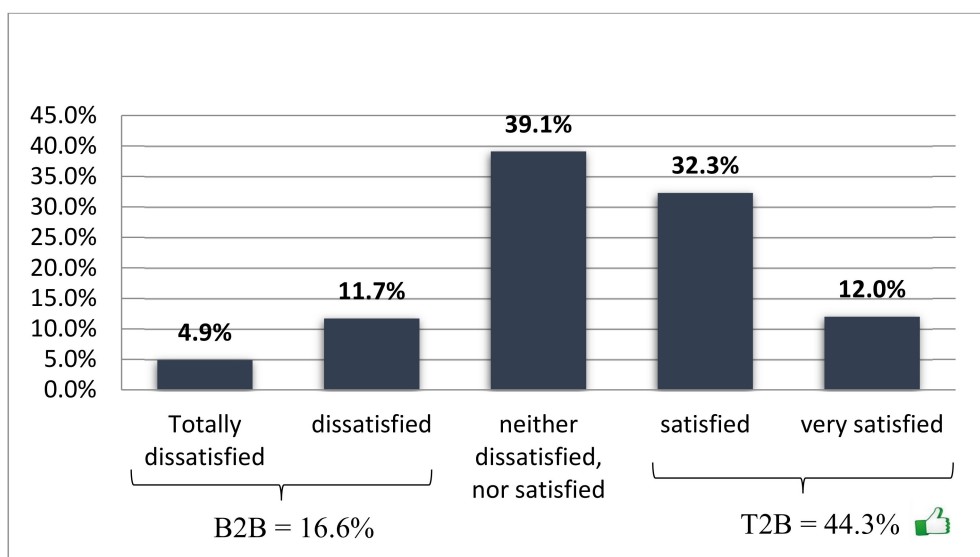

**Figure 2.** Frequency of responses regarding respondents' degree of satisfaction with online services offered by public utility institutions and companies.

This research aimed to establish the areas in which respondents felt that public authorities in their city should invest more in technology.

Most of the answers emphasized the field of health as being the one that requires the most investments regarding technology, which was mentioned by almost half of the respondents (42%). A total of 29% of the respondents mentioned protection of the environment, followed by 23% who opted for the field of education. The lowest percentage, 5%, was given for transportation, while 1% selected the "another field" answer.

During the pandemic, health system shortcomings emerged, especially at the national level, which included technology. Similarly, education suffered a lot during the COVID-19 period, and it was decided that by "order of the Minister of Education and Research, based on the decision of the National Committee for Emergency Situations and the analysis of the epidemiological situation at national level, the suspension of the activities that require physical presence can be decided [ . . . ] in the educational units and the continuation of the didactic activities in the online system" [84]. This decision revealed the shortcomings in the educational system. Unfortunately, the repercussions of these decisions will be seen in the coming years.

One thing is certain, and it has been highlighted once again by this research, namely that the population wants major investments in technology in the fields of health, environmental protection and education.

As a result of citizens' desire and the administrations' need for technology, the Authority for the Digitization of Romania was established in 2020, which aims to digitize all government and administrative activities on a large scale [85].

The connection between the use of online services offered by the public utilities institutions and companies and the income of respondents was analyzed, with the following table showing the results obtained.

The authors notice from the data analysis (Table 5) that the distribution of respondents using the online services provided by the public utilities institutions and companies is significantly higher for higher-income earners. It is noticed that the highest percentage (96%) was recorded for people who earn more than RON 5000. The share remains at a high level (82%) for those earning between RON 3001 and 5000.

**Table 5.** The contingency table regarding the use of online services provided by public utilities institutions and companies in relation to the income of respondents.

| | | | Net Monthly Income Is in the Range of: | | | | |
|---|---|---|---|---|---|---|---|
| | | | Less than RON 1.400 | RON 1.401–3.000 | RON 3.001–5.000 | More than RON 5.000 | Total |
| Use the online services | No | Count | 20 | 80 | 72 | 4 | 176 |
| | | % | 16.1% | 16.7% | 17.6% | 3.8% | 15.8% |
| | Yes | Count | 104 | 400 | 336 | 100 | 940 |
| | | % | 83.9% | 83.3% | 82.4% | 96.2% | 84.2% |
| | Total | Count | 124 | 480 | 408 | 104 | 1116 |
| | | % | 100% | 100% | 100% | 100% | 100% |

At the other end of the spectrum, less high-income respondents (over RON 5000) responded that they had not used such online services so far, amounting to 3.8%.

In conclusion, it is noted that, from among the researched population, those with a higher income use the online services offered by the public utility companies to a higher degree.

*O3. Finding out the most important actions in **renewable energy***

From the respondents, 41% have currently had a smart meter installed in the building where they live, while 39% have not had anything installed and a relatively high percentage of 20% of the respondents answered that they did not know.

Most respondents, namely 89%, do not use renewable energy sources, while only 11% use them. The percentage of those who use such energy sources will most likely increase in the coming period, a statement based on the European Union's commitment to meet the goal that at least 27% of total energy consumption should be energy from renewable sources by 2030 [86]. In this research, the respondents who mentioned that they used renewable energy were asked to mention the type they used. The research results showed that 93% of the respondents who used renewable energy used geothermal energy, while 7% used solar energy.

A study carried out in Poland showed that the most used alternative source of energy in the country was solar (64%), followed by biogas (18%), hydropower (8%), biomass (7%) and wind (3%) [52]. We can say that, depending on natural resources and geographical positioning, each country/city tries to reap the benefits of using the most efficient renewable energy sources.

*O4. Identifying **smart waste management** actions*

Taking into consideration that the norms of the European Union oblige Romania to reach a degree of waste recycling of at least 50% by 2020 [87], the results of the research highlight a major problem, namely the fact that there are very low percentages of people who recycle waste. According to the results, the majority of the respondents (61.3%) do not select the waste to be disposed of, while only 38.7% of the respondents select them.

The respondents who do not select waste when dis-posing of it were asked to mention the reasons why they do not select waste. For this question, the respondents had the opportunity to tick several answers.

The absence of containers for waste selection is mentioned as being the most representative reason for why respondents do not select waste before disposing of it, with a majority of respondents (91.8%) selecting this answer. A relatively low percentage (15.2%) of the respondents mentioned convenience, while 2.3% had other reasons. The lowest percentage (0.6%) represents the "I am not interested" answer.



The authors used the chi-square test to check whether there was a link between the gender of respondents and their habit of selecting household waste. The expected values were calculated (Table 6).

**Table 6.** The observed and expected frequencies.

| | | | Gender | | |
|---|---|---|---|---|---|
| | | | Female | Male | Total |
| **The habit of selecting waste** | No | Count | 368 | 316 | 684 |
| | | Expected Count | 34,813 | 33,587 | 684 |
| | Yes | Count | 200 | 232 | 432 |
| | | Expected Count | 21,987 | 21,213 | 432 |
| | Total | Count | 568 | 548 | 1116 |
| | | Expected Count | 568 | 548 | 1116 |

The expected frequencies are on the expected count line. From the summary analysis of the differences between the absolute frequencies (Count) and the expected ones (Expected Count), there are differences at the level of all subgroups.

For women, the expected values for waste selection are higher than the absolute values. For men, the expected values are lower than the absolute ones. For those who do not select waste before disposing of it, women have a higher value for absolute values than expected values, whereas for men the expected values are higher than the absolute ones. In order to test the significance of the differences, chi-square tests were used (Table 7).

**Table 7.** Critical report for the chi-square analysis.

| Chi-Square Tests | | | |
|---|---|---|---|
| | **Value** | **df** | **Asymp. Sig. (2-Sided)** |
| Pearson Chi-Square | 5.967080238 | 1 | 0.014575395 |
| Likelihood Ratio | 5.971133884 | 1 | 0.014541927 |
| Linear-by-Linear Association | 5.961733392 | 1 | 0.014619661 |
| N of Valid Cases | 1116 | | 0.014575395 |

a Computed only for a 2 × 2 table; b 0 cells (0%) have an expected count less than 5. The minimum expected count is 21,213.

Applying the chi-square test and comparing the calculated level of significance (0.014) with the theoretical one (0.05), it is noticed that the first level of significance is lower. Thus, we conclude that there is a connection between the gender of the researched population and the habit of selecting waste before disposing of it.

In conclusion, the quantitative research carried out has achieved its purpose and objectives, and the obtained results are really useful. Objectives 1 (determining smart transportation practices, including public and private transportation), 2 (identifying smart governance practices, including online administrative services provided by public institutions and companies) and 4 (identifying the smart waste management actions) were almost completely achieved. Objective 3 (finding out the most important actions in renewable energy) was achieved to a high extent. Regarding the research hypotheses, the situation is as follows: Hypothesis 1 (the majority of respondents know the Smart City concept) was disproved as the research showed that only 47% of the subjects knew of this concept; Hypothesis 2 (a large part of respondents do not use renewable energy sources) was proven to be true as 89% of respondents did not use renewable energy sources; and Hypothesis 3 (most subjects selectively collect waste) was disproved as the responses revealed that only 38.7% of the respondents selectively collected waste.

A summary of the most relevant research conclusions obtained from the quantitative research on the four main objectives is given in Figure 3.

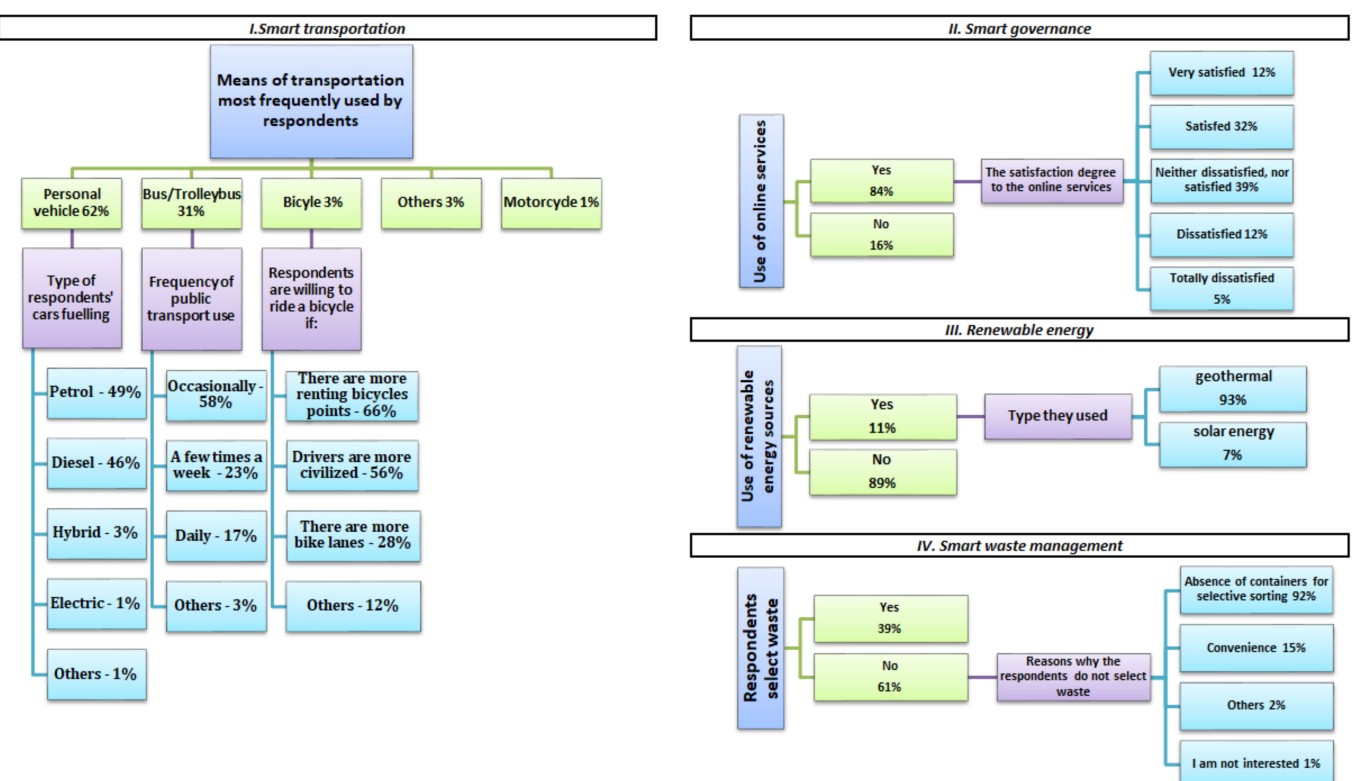

**Figure 3.** Framework of the most relevant research conclusions.

## 5. Conclusions

This paper highlighted the fact that the concept of Smart City has experienced a wide development in recent years, including in Romania where important efforts have been made to develop the sustainability of cities by implementing smart measures in transportation, renewable energy, e-governance and waste management. This study enriches the literature and research that has been conducted in the Smart City field. The Smart City concept has emerged in the last decade, but a new concept, that of a sustainable Smart City, is already under discussion. First of all, this paper has been able to identify and group the four key features of a sustainable Smart City: smart transportation, smart governance, smart waste management and renewable energy.

The article is original through the two types of research performed. The first is a comparative analysis, and the second is quantitative research. The quantitative research conducted in four cities in the country (Brașov, Cluj-Napoca, Sibiu, Bucharest) showed that over half of the respondents are familiar with the Smart City concept and 41.9% consider health as a priority for technological investment.

Considering all the aspects mentioned and analyzed in this article, the authors of propose clearer highlighting and division of cities from the Smart City point of view. The authors' proposal is to *calculate an economic and sustainable index/indicator* for each city, which measures how smart a city is. For this, an analysis of the four characteristics of a Smart City (identified as the key features of a sustainable Smart City) must be performed and, depending on the value of the *index/indicator*, the city must be included in one of the four levels of achievement proposed by the authors:

- Indexed city "Sustainable Smart City" in *a majority* global percentage;
- Indexed city "Sustainable Smart City" in *an average/medium* percentage;
- Indexed city "Sustainable Smart City" in *a small* percentage;
- *NON* "Sustainable Smart City".

The proposal of the authors to represent the Smart City fulfillment stage is represented in Figure 4.

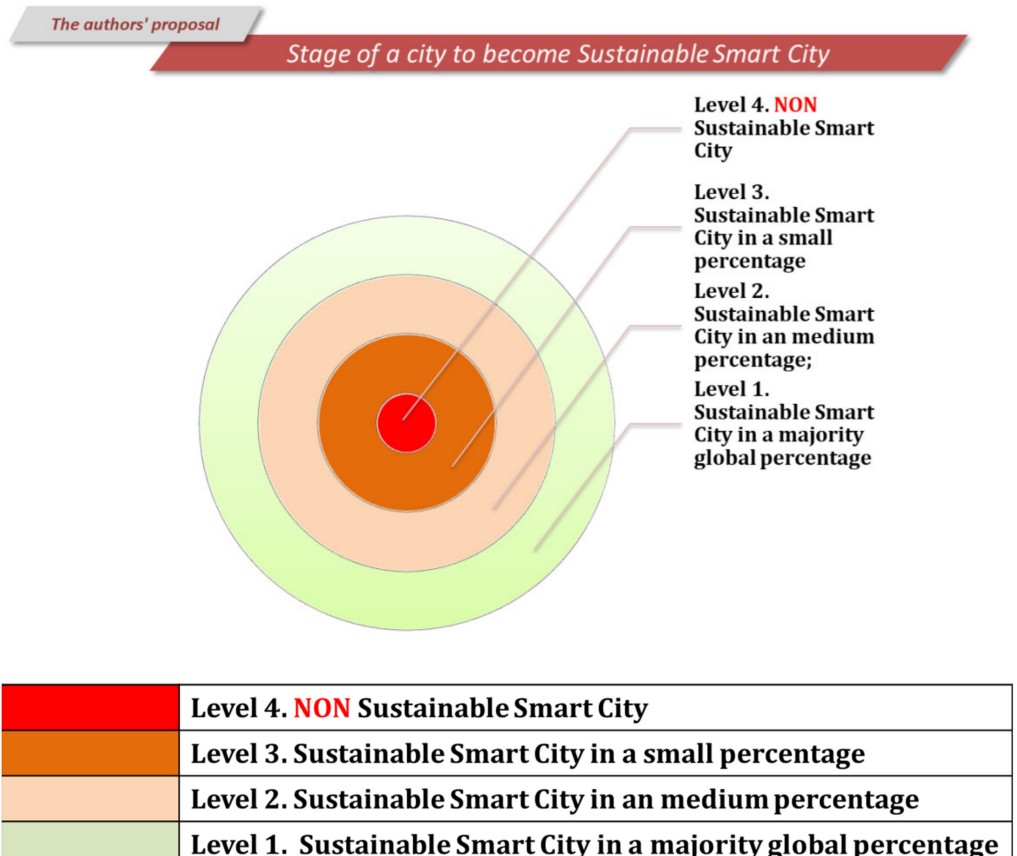

**Figure 4.** The proposal of the authors to represent Smart City stages.

This method of calculation and representation can be applied by any city. In the next period, it becomes important to perform this analysis in order to see how the cities are at the specified time. This could be a zero moment. Of course, this model may represent the analysis of a future article.

The "Smart City Solutions for a Riskier World" study shows that while the COVID-19pandemic has created very big problems and obstacles throughout the world, it has also created a wave of innovation. This study includes research conducted between August and September 2020 based on a survey of senior officials from 167 cities in 82 countries around the world. Cities were assessed and ranked, based on progress, into two categories: progress in applying smart solutions, with cities ranked as "starter", "intermediate" or "leader"; and progress regarding the United Nations Sustainable Development Goals (SDGs), with cities classified as "implementer", "developer" or "sprinter". ESI ThoughtLab research, sponsored by Oracle, Deloitte and Intel, highlights the vital role technology, data, cyber security and public–private partnerships play in ensuring a healthy, safe and prosperous future for citizens after this pandemic. The study shows that the COVID-19 pandemic is accelerating innovation in the public sector; 88% of surveyed city leaders are calling for investment in cloud platforms. In the study, Bucharest is classified as "evolving" according to the United Nations in terms of the Sustainable Development Goals and, based on the use of digital solutions and technologies to achieve social, environmental and economic goals, it is classified as "intermediate" [88].

The limitation of the quantitative research carried out in this paper is the fact that it was not possible to carry out a non-random sampling, though this aspect does not mean that the research is less valuable. Another limitation of the research is that only four cities

in Romania were analyzed. In the future, the authors are considering conducting more extensive research, which will include more cities from Europe.

The current global energy crisis, generated by the war in Ukraine, as well as increases in the price of energy resources (oil, refined products, gasoline, diesel, gas, liquefied gas, electricity, etc.) can be considered from this point of view as an opportunity, the key to a faster transition to a green economy and society, especially in the EU space. There is a favourable context determined by the adoption, at the EU level, of the National Recovery and Resilience Plan [89], which has in its structure several components aimed at the support given to member countries to move to a green economy and society. Taking into account these aspects, we can state that there are two major directions of action in the public authority–citizen relationship, namely stimulation and coercion, in the sense of adopting the measures proposed by this plan. Thus, we highlighted the measures that mainly involve the stimulation of citizens to participate in the implementation of green environmental policies, challenges and recommendations regarding the four objectives of the research.

Regarding *smart public transport*, in the context of the increase in air pollution in big cities, pollution mainly due to vehicles, we recommend replacing the car fleet with electric cars. The countries of the world must create financial aid schemes for citizens and companies which will lead to the replacement of thermal engines with non-polluting electric cars. In this sense, good practice models in this field should be taken into account. Romania managed to have a more than four-fold increase in the number of electric cars. This increase is mainly due to the Rabla+ Program, which envisages receiving Ecobonuses of over EUR 10,000 when buying an electric car [90].

For a city to become a Smart City, the *digital transformation* component represents a new vision in the public sector.

The advantages of digital transformation are efficiency, transparency and simplicity, and these lead to much higher productivity of processes. The awareness of the need for the introduction of new technologies by the leaders of public institutions, the continuous adaptation to the demands of citizens and the provision of quality, safe and fast online public services are just some of the vision elements that contribute to the development of the Smart City concept.

In the context of increases in pollution of all types and climate change, the recommendations take into account *public management* at the central and local level. Here, we refer to all the programs offered by the Romanian Ministry of Environment, Water and Forests. These programs are applied regionally and locally [91].

In the component related to *waste management*, in the context of increasing waste pollution, we refer to the way some public policies have been adopted by regional or municipal authorities. An example in this case can be represented by the mode of action of a regional authority in Romania (Brasov County Council). This is an example of good practice for adopting the European Union's waste management policy [92].

Other recommended measures could aim at better communication and educating citizens about household waste collection. An example of this can be found in the city of Brasov, Romania, where PETrica collection machines were introduced. These machines allow the efficient collection of plastic, glass and aluminum waste. Citizens who collect are rewarded with free tickets for local public transport [93].

The research carried out and presented in this article is topical and, for our country, the presented information is of real use. The results of the quantitative research add value to the targeted cities, managing to create a base from which to start the whole Smart City mechanism.

The concept, and especially the philosophy, of a Smart City presupposes the transition from a passive consumer of resources (transport, infrastructure, health, education, etc.) to a prosumer, a person who creates resources (by producing more than they consume). A future research direction could study the relationship between the Smart City concept and that of a circular economy. For each urban community that aspires to Smart City status,

designing public decentralized policies and local autonomy that will lead to the creation of a competitive attitude is necessary.

The authors expect that, after completing this work, they will enrich the specialized literature through a better knowledge of developments in Romania on the topic of this work. In addition, this work can be the basis for the definition of public policies by regional and local authorities on the four key Smart City components. Furthermore, the information and research from this work can help the assimilation, by local authorities in Romanian cities, of some models of good practice in the fields of communication, information and education for all parties involved in Smart City processes. The Green Cities Forum is an example, being the largest event in Romania dedicated to environmental sustainability [94]. The adoption of a system of indicators (as proposed in this paper) that measures how intelligent a city is, would first allow all cities to understand their position in the implementation/development of the city as a Smart City. Secondly, it would help entrepreneurs decide where to invest. Last but not least, this system would give citizens the opportunity to decide on their own quality of life.

Another research direction could include the evaluation of current public policies for urban development and their correlation with the concept and components of a Smart City. Additionally, there are already studies and articles that focus on energy supply, which is considered as an active research topic among the new aspects of urban management, especially in developing countries [95].

**Author Contributions:** Conceptualization, S.B., N.A.N., A.M., A.Z. and M.B.; methodology, N.A.N., S.B. and A.Z.; literature review, A.M. and S.B.; analysis and writing the results, A.Z. and N.A.N.; results and discussion, S.B., N.A.N., A.M. and A.Z.; conclusions, S.B., N.A.N., A.M. and A.Z; writing—original draft preparation, S.B., N.A.N., A.M., A.Z. and M.B. All authors have read and agreed to the published version of the manuscript.

**Funding:** This research was funded by TRANSILVANIA UNIVERSITY OF BRAȘOV.

**Institutional Review Board Statement:** Ethical review and approval were waived for this study due to we conducted a non-interventional study (questionnaires type) and all participants were fully informed that the anonymity is assured. All the participants agreed to answer the questions, taking into account that the data obtained were used strictly for statistics.

**Informed Consent Statement:** Informed consent was obtained from all subjects involved in the study.

**Data Availability Statement:** Not applicable.

**Conflicts of Interest:** The authors declare no conflict of interest.

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
