# Peer review of "Research of the Smart City Concept in Romanian Cities"

_sustainability, doi:10.3390/su141610004_

Round 1

Reviewer 1 Report

In general, the article seems a bit confusing to me. First of all, the title is not good, the comma should be removed and reworded. In theory, the smart city already should be inherently sustainable

The background should be better explained in the Introduction. What you wrote in lines 82-90 should be merged with the Introduction, as well as paragraph 2 which turns out to be a mix of many concepts. You said you did a literature review to identify "key features." First of all, it is not clear how the literature was selected. Second, models already exist for defining "key features" (look up the various pillars of smart cities, such as the Smart City Ranking theorized by the University of Vienna and Ljubljana, but there are many others)

The results are somewhat generic, and should be better referenced through comparison with other literature studies, as well as future challenges and recommendations should be highlighted

MINOR

Line 376. Acronyms should be written after the full word, not the other way around

Bibliographies should be written like this: Author et al. (xxxx) [xx]. The year is missing.

Remove boldface from titles (although the editorial board will eventually take care of this).

Author Response

Response to Reviewer 1 Comments

We are thankful for the comments as they are extremely useful to improve the quality of our article.

Point 1: The title is not good; the comma should be removed and reworded. In theory, the smart city already should be inherently sustainable.

Response 1: The authors changed the title according to the recommendations received from reviewer 1 and 3. They removed the comma and the word sustainability, and added that the study refers to cities in Romania.

Point 2: The background should be better explained in the introduction. What you wrote in lines 82-90 should be merged with the Introduction, as well as paragraph 2 which turns out to be a mix of many concepts. You said you did a literature review to identify key features. First of all, it is not clear how the literature was selected. Second, models already exist.

Response 2: The paragraph on lines 82-90 has been moved to the beginning of the introduction section. The paragraph at lines 100-107 was moved after the previous one, as the authors thought it more appropriate to the context.

Between lines 53-54 a paragraph about how the literature was selected for this work was added.

The authors changed from “identify” key features to “take into account” these features.

Point 3: The results should be better referenced through comparison with other studies; as well the future challenges and recommendations should be highlighted.

Response 3: The results were supplemented by adding two bibliographic sources for comparison, but the authors did not find other studies/researches on the topic considered in the quantitative research.

The future challenges and recommendations of the authors have been highlighted in the conclusion section.

Point 4: Acronyms should be written after the full word

Response 4: All acronyms in the paper have been moved after full words.

Point 5: Bibliographies should be written like this: Author et al. (xxxx)[xx]

Response 5: The authors wrote the bibliography in accordance with the journal's requirements.

Reviewer 2 Report

Comments and Suggestions for Authors

1. Abstract

1.1. The Abstract contains the necessary elements: article proposal, methodology, and main results. 

2. Introduction  

2.1. Lines 62 to 864 – Please, inform the four key features. Authors should explain the reasons that led them to focus on only one of the four key features. It's okay to focus on just one key feature, but it should be explained why.

2.2. The relevance/contribution of the work needs to be better presented.

2.3. At the end of the Introduction, please add information about the paper structure and a short description of its sections.

3. Body part of the paper

3.1. In the Illustrations produced by the authors, the source must not be informed. It is implied that the illustrations derive from the authors' knowledge from the research performed (for example, "Source: Made by the authors, based on the results of research," "Source: Author's calculation based on collected data."

3.2. The section "3. Research methodology" needs to be expanded. The bibliographic research, the participant selection procedure, and the sample characteristics are essential parts of the methodology. The authors should address this comment carefully so that readers can conclude that the manuscript is based on a solid methodology and its results are robust.

a) In lines 168 – 169 - "This paper, based on the study of literature, identified several elements of a Smart city." The authors state that they performed a literature review. The procedures for selecting the most relevant works must be presented. The authors should explain with which combination of keywords and how many articles they were evaluated, and with which criteria and how many articles were excluded. Please see the PRISMA guidelines for a proper understanding of my recommendation.

b) Authors need to provide more information about the survey. How were the respondents selected? How were they invited? How do you ensure that the sample is representative? This part should be rewritten to include a detailed description of the participant selection procedure, the specific criteria based on which they were selected, and how they can guarantee the validity of their results.

3.3. Line 387 - Table 2 is a result of the survey (for example, the authors did not previously determine that they would interview 49% of men.) and should be moved to the appropriate section.

3.4. A more detailed description of how the authors expect this work to impact stakeholders might improve the discussion significantly. 

4. Conclusions

4.1. Line 162 - Please remove "The authors of."

4.2. The authors should better highlight the relevance of the results.

Author Response

Response to Reviewer 2 Comments

We greatly appreciate the reviewer’s very constructive and thoughtful comments.  Below we provide responses to each comment along with the associated changes we made to the manuscript. This report summarizes our responses to all the comments.

Point 1: Lines 62 to 864 – Please, inform the four key features. Authors should explain the reasons that led them to focus on only one of the four key features. It's okay to focus on just one key feature, but it should be explained why.

Response: It was further specified from row 62 what are the reasons for choosing only 4 of the components of a Smart City.

Point 2: The relevance/contribution of the work needs to be better presented.

Response:  The authors included this requirement in the last paragraph of the Introduction.

Point 3: At the end of the Introduction, please add information about the paper structure and a short description of its sections.

Response: As you suggested, at the end of the introduction we made a summary of the article and a brief description of each section.

Point 4:  In the Illustrations produced by the authors, the source must not be informed. It is implied that the illustrations derive from the authors' knowledge from the research performed (for example, "Source: Made by the authors, based on the results of research," "Source: Author's calculation based on collected data."

Response: They were moved all the statements regarding "Source: Made by the authors, based on the results of research," "Source: Author's calculation based on collected data."

Point 5: The section "3. Research methodology" needs to be expanded. The bibliographic research, the participant selection procedure, and the sample characteristics are essential parts of the methodology. The authors should address this comment carefully so that readers can conclude that the manuscript is based on a solid methodology and its results are robust.

  1. In lines 168 – 169 - "This paper, based on the study of literature, identified several elements of a Smart city." The authors state that they performed a literature review. The procedures for selecting the most relevant works must be presented. The authors should explain with which combination of keywords and how many articles they were evaluated, and with which criteria and how many articles were excluded. Please see the PRISMA guidelines for a proper understanding of my recommendation.

Response: We added the selection procedure of the most relevant scientific articles studied.

  1. Authors need to provide more information about the survey. How were the respondents selected? How were they invited? How do you ensure that the sample is representative? This part should be rewritten to include a detailed description of the participant selection procedure, the specific criteria based on which they were selected, and how they can guarantee the validity of their results.

Response: In section 3 Research Methodology we added explanations for the sampling and we explained that, being a non-random sampling method, the results cannot be extrapolated. The authors resorted to this sampling technique (snowballs) in order to obtain the highest possible response rate.

“The sampling method used in this research is the one called "snowball". It is a non-probability sampling technique that does not allow extrapolation of results. The purpose of this sampling method is to get as many people as possible recruited into the sample. In this sense, the link leading to the web page of the questionnaire was distributed on social networks (Facebook, WhatsApp, Instagram) and the users of these networks were asked to access the questionnaire and answer its questions. Two eligibility criteria were taken into account: being over 18 years old and living in one of the four analyzed cities (Brașov, Bucharest, Cluj and Sibiu). “

Point 6: Line 387 – Table 2 is a result of the survey (for example, the authors did not previously determine that they would interview 49% of men.) and should be moved to the appropriate section.

Response: Table 2, regarding the structure of the sample, was moved to section 4.2 and became table 3.

Point 7: A more detailed description of how the authors expect this work to impact stakeholders might improve the discussion significantly. 

Response: At the end of the conclusions section, in the penultimate paragraph, the authors have added a more detailed description of how this work is expected to affect stakeholders.

Point 8: Line 162 - Please remove "The authors of."

Response:  The authors have corrected, according to your suggestion.

Point 9: The authors should better highlight the relevance of the results.

Response: The relevance of the results have been highlighted in the conclusion section.

Reviewer 3 Report

Thank you to the authors for a very interesting article.

In today's world, the role of cities as major centres of life bringing together an increasing proportion of the population is growing. Consequently, urban authorities are faced with the necessity of guaranteeing an adequate standard and living conditions for their inhabitants. I therefore believe that the authors present a very important problem.

My comments/suggestions for change:

1/ I believe that the Authors should present the benefits and risks of implementing the smart city concept in the studied cities;

2/ The authors only define the objectives of the study. In the reviewer's opinion the research hypotheses are missing;

3/ in the conclusion, please state whether the objectives of the research have been achieved and to what extent;

4/ in the title, please add that the article is about cities in Romania.

Author Response

Response to Reviewer 3 Comments

We greatly appreciate the reviewer’s very constructive and thoughtful comments.  Below we provide responses to each comment along with the associated changes we made to the manuscript. We have thoroughly studied them and revised the manuscript accordingly. This report summarizes our responses to all the comments.

Point 1/ I believe that the Authors should present the benefits and risks of implementing the smart city concept in the studied cities;

Response: At the end of section 4.1, the authors presented the benefits and risks of implementing the smart city concept in the studied cities.

Point 2/ The authors only define the objectives of the study. In the reviewer's opinion the research hypotheses are missing;

Response: In the second paragraph of the Research Methodology, the authors have specified more clearly what the general hypotheses of this quantitative research are.

Point 3/ in the conclusion, please state whether the objectives of the research have been achieved and to what extent;

Response: In the conclusions section, the authors specified whether the proposed objectives were achieved.

Point 4/ in the title, please add that the article is about cities in Romania.

Response: The authors changed the title according to the recommendations received from reviewer 1 and 3.

Round 2

Reviewer 1 Report

Revisions have been properly implemented.

Reviewer 2 Report

Reviewer Comments to Author

Dear authors,

Thank you for addressing my comments.

With kind regards,